# A Probabilistic Structural Equation Model to Evaluate Links between Gut Microbiota and Body Weights of Chicken Fed or Not Fed Insect Larvae

**DOI:** 10.3390/biology11030357

**Published:** 2022-02-23

**Authors:** Johann Detilleux, Nassim Moula, Edwin Dawans, Bernard Taminiau, Georges Daube, Pascal Leroy

**Affiliations:** Fundamental and Applied Research for Animals, Veterinary Faculty, University of Liège, Quartier Vallée 2, Avenue de Cureghem 7A-7D, 4000 Liège, Belgium; nassim.moula@uliege.be (N.M.); edawans@uliege.be (E.D.); bernard.taminiau@uliege.be (B.T.); georges.daube@uliege.be (G.D.); pascal.leroy@uliege.be (P.L.)

**Keywords:** microbiota, 16S RNA, Bayesian network, structural equation model, chicken, insect in feed, black soldier fly

## Abstract

**Simple Summary:**

Feeding poultry with insects could reduce production costs, but the impact of this diet on their gut microbiota and growth is little known because the network of relationships between their weights, the composition of their microbiota and their diet is complex and potentially biased by confounding factors (such as the gut compartment, age and sex of the birds). In this study, we were able to unravel these relationships in local breed chickens fed or not fed with black soldier fly larvae thanks to a technique of artificial intelligence (the probabilistic structural equation model). Bacteria were grouped into few entities with distinctive metabolic attributes and were probably linked nutritionally. Birds’ age influenced body weights and bacterial composition. The proposed methodology was thus able to simplify the complex dependencies among bacteria present in the gut and to highlight links potentially important in the response of chicken to insect feed.

**Abstract:**

Feeding chicken with black soldier fly larvae (BSF) may influence their rates of growth via effects on the composition of their gut microbiota. To verify this hypothesis, we aim to evaluate a probabilistic structural equation model because it can unravel the complex web of relationships that exist between the bacteria involved in digestion and evaluate whether these influence bird growth. We followed 90 chickens fed diets supplemented with 0%, 5% or 10% BSF and measured the strength of the relationship between their weight and the relative abundance of bacteria (OTU) present in their cecum or cloaca at 16, 28, 39, 67 or 73 days of age, while adjusting for potential confounding effects of their age and sex. Results showed that OTUs (62 genera) could be combined into ten latent constructs with distinctive metabolic attributes. Links were discovered between these constructs that suggest nutritional relationships. Age directly influenced weights and microbiotal composition, and three constructs indirectly influenced weights via their dependencies on age. The proposed methodology was able to simplify dependencies among OTUs into knowledgeable constructs and to highlight links potentially important to understand the role of insect feed and of microbiota in chicken growth.

## 1. Introduction

The use of insects in poultry feed has generated great interest based on the recognition that they are powerful bio-converters with a low environmental footprint [1]. Insects are naturally consumed by poultry and, when supplemented in basic diets, may improve poultry growth performances [2]. These performances depend on the multiple interactions between feed components, host cells and gut microbiota: When gut microbiota is modified, digestion and nutrient absorption are affected. These, in turn, can influence feed conversion, leading to altered growth and health [3].

The microbiota in gut mucus and lumen samples plays a role in the metabolism of chickens [4]. Indeed, bacterial fermentation of foodstuff entering the caeca produces up to 10% of the metabolizable energy [5]. The caecum is also the gastrointestinal tract compartment where the number and variety of bacteria are the highest, with up to 10^11^ bacteria per gram [6]. The composition of the chicken caecal bacterial flora has been studied by numerous authors, and significant variations were found according to the age and genetics of the birds [7], their surrounding environment [8] and diet composition [9]. Regarding the impact of insects, researchers found differences in the microbiota of broilers fed basic feed supplemented with 0% to 15% of partially defatted black soldier fly (BSF) larvae [10], in the microbiota of hens fed meals in which soybean was completely or not replaced by defatted BSF larvae [11] and in the microbiota of broiler Ross 708 chickens fed meals supplemented with 0% to 15% of Tenebrio Molitor [12]. Inversely, no differences were found in the composition of the caecal microbiota of broilers fed a basic diet supplemented with 0% to 8% of defrosted BSF larvae [13].

The different nutrients in BSF larvae can influence the composition of the broiler gut microbiota because these microbes use feed-derived compounds as growth substrates [14]. Although the content of BSF larvae varies considerably depending on their rearing substrate, the developmental life stage at which they are harvested and the method used to kill them, it is generally rich in protein (dry matter > 30%) and fat (dry matter > 20%), with an abundance of lauric acid known for its antimicrobial activity against Gram-positive bacteria [15]. It was also reported that chitin, the main component of the exoskeleton, serves as a fermentation source for caecal microbiota [16].

The analysis of the gut microbiotal composition may also vary with the methods used for their quantification, whether it is in the sequencing [17] or in the statistical methods of analysis. Many classical statistical tests are available to analyze the gut microbiome [18] but it is important to recognize that the evaluation of processed metagenomic data have several characteristics that necessitate specific statistical tools. Typically, these data are normalized into non-negative relative abundances of operational taxonomic units (OTU) [19], and because they are fundamentally discrete, these can only be approximately modeled by continuous variables [20]. Relative abundances are by nature the estimates of the multinomial probabilities for the OTU counts. These have compositional characteristics [19,21] and must be log-transformed to resolve the constant sum constraint [22]. Moreover, distributions of relative abundances are often highly skewed (over-dispersed) with a lot of zero values [23]. Datasets are usually very large, with the number of sampled animals being much lower than the number of bacterial species (“big-p, little-n” problem). In such a case, the probability of finding an OTU that correlates with the variable of interest is high, even if no real correlation exists in the domain. Another issue comes from the interaction within and between microbial communities [24] and the potential dependencies between OTU relative abundances (multicollinearity), which prevents the use of traditional regression methods to test whether an association exists between relative abundances and a variable of interest [25].

To address these issues, networks have been proposed to describe the relationships between OTU relative abundances [26,27], including Bayesian networks [28]. Bayesian networks are represented by directed acyclic graphs in which variables are symbolized by connected nodes: If a node depends on another node then a directed edge is drawn between them (direct probabilistic dependencies). A node is interpreted as the child node if it has the tip of the arrow attached to it, and as the parent node if it has the base of the arrow attached to it. Each node is associated with a probability distribution that indicates how the probability that a child node will take a certain value depends on the values taken by its parent node. Different algorithms exist to construct the structure of a network (i.e., the description of the dependence/independence relationships among nodes) from the data [29]. One method is to select the network that has the best fit to the data. This fit can be measured by various parameters, including the minimum description length score function (MDL) as a stopping rule. The MDL function is characterized by a preference for simpler Bayesian networks than complex ones: a network with a lower MDL score indicates that it optimizes the tradeoff between complexity and fit more effectively than one with a higher score [30]. Once the structure of the network is known, conditional probability tables can be estimated by maximizing their likelihoods [31]. The MDL function can also be used to estimate the number of classes when discretizing OTU relative abundances with minimal information loss and maximum accuracy in representing the abundances, as it is required in most methods available to learn networks from data (e.g., BayesiaLab, Hugin Expert, Netica, R).

Besides Bayesian networks, structural equation models (SEMs) are another type of graphical model, which are useful to analyze the potential relationships between OTU relative abundances and a phenotype of interest [32,33]. These models allow for grouping relative abundances into a limited number of latent constructs (LC), which represent the underlying unobservable causes of variations in the observed abundances. Commonly, SEMs are composed of measurement and structural models. The measurement model describes the relationships between observed data and LCs while the structural model imputes the relationships between LCs and a variable of interest [34,35]. In the measurement part of the SEM, relationships are often based on known or expected underlying mechanisms. However, in probabilistic SEM (pSEM), they are inferred assuming no theory but based on the structure of a network [36]. In the structural part of the SEM, it is also possible to break down the relationship between a LC and a variable of interest into its direct and indirect components. In the terminology of causal mediation analysis, a “direct effect” is an effect that is not mediated by other variables and an “indirect effect” is the portion of the effect that can be explained by mediation alone. Their sum is sometimes called the ‘total effect’ [37,38].

The goals of this study are, using the pSEM methodology: (1) to summarize into a few LCs all relationships observed between relative abundances of OTUs found in the caeca and cloacal swabs of chickens fed or not fed BSF larvae, and (2) to estimate the effects (adjusted for potential confounders) of the presence of these larvae on the weights measured on the day of OTU collection, with effects mediated or not by changes in the microbiota in the LCs.

## 2. Materials and Methods

### 2.1. Bird and Insect Management

Management of insect and birds was kept intentionally simple, in semi-artificial conditions. The BSF larvae were raised on beet pulp, which was watered manually so the substrate remained more or less “sludgy”, and under ambient temperature maintained between 20 and 30 °C. The larvae were collected by hand before the pupal stage and kept at −20 °C directly after collection and until bird feeding time. A portion of the larvae was allowed to complete development to adulthood. Adult flies were transferred into cages with natural and artificial lighting for mating and egg-laying [39].

The experiment consisted in raising thirty one-day-old chicks of the local breed Ardennaise indoors at the Veterinary Faculty of Liege, all together for the first days of life to ensure microbiota exchange through typical bird behavior [40]. At the end of this “pre-experimental” period, we collected fecal samples via cloacal swabs and randomly allocated birds to three groups with ten birds per group and allowed them to adjust over two days. During this “experimental period”, birds had ad libitum access to water and to commercial feed supplemented with either 0%, 5% or 10% BSF larvae. Appendix A provides information about the ingredients and chemical composition of the diets, which were formulated to be of equal energy and protein concentrations. Birds were weighed on a weekly basis until the end of the experimental period, when fecal samples were collected from the caecum after birds were euthanized by cervical dislocation. The physical environment was identical for all three groups and the whole experiment was replicated three times. The sampling frame and age at slaughter were a little different across replicates because of external conditions related to the sanitary containment of COVID-19: In the first replicate, we collected 5 cloacal (28 days of age) and 5 caecal (67 days of age) contents per group. In the second replicate, we collected 4 cloacal (16 days of age) and 8 caecal (73 days of age) contents per group. In the last replicate, we collected 8 caecal contents per group at 39 days of age. Contents of the caecum and cloacae were placed individually in sterile tubes with phosphate-buffered saline and stored immediately after collection at −80 °C until DNA extraction.

### 2.2. 16S rDNA Detection and Analyses

Total bacterial DNA was extracted from the samples with the PSP Spin Stool DNA Plus Kit 00310 (Invitek, Berlin, Germany), following the manufacturer’s recommendations. The PCR amplification of the 16S rDNA V1–V3 hypervariable region and library preparation was performed with the following primers (with Illumina overhand adapters): forward (50-GAGAGTTTGATYMTGGCTCAG-30) and reverse (50-ACCGCGGCTGCTGGCAC-30). Each PCR product was purified with the Agencourt AMPure XP bead kit (Beckman Coulter, Pasadena, CA, USA) and submitted to a second PCR round for indexing, using Nextera XT index primers 1 and 2. After purification, PCR products were quantified using the Quant-IT PicoGreen (ThermoFisher Scientific; Waltham, MA, USA) and diluted to 10 ng/μL. A final quantification of each library was performed using the KAPA SYBR^®^ FAST qPCR Kit (KapaBiosystems; Wilmington, MA, USA) before normalization, pooling and sequencing on a MiSeq sequencer using V3 reagents (Illumina; San Diego, CA, USA).

Sequence read processing was performed as previously described using the MOTHUR software package v1.39.5 [41] and the VSEARCH algorithm for chimera detection [42]. A clustering distance of 0.03 was used for OTU generation. The 16S reference alignment and taxonomical assignment from phylum to genus were performed with MOTHUR and were based upon the SILVA database (v1.32) of full-length 16S rDNA sequences [43]. Subsample datasets with 10,000 cleaned reads per sample were obtained and used to evaluate all OTUs’ coverage (Good’s coverage), which was adequate.

### 2.3. Data Processing and Modeling

For each individual, relative abundances were computed as the number of OTUs of one genus divided by the total number of OTUs. As an editing step, the dataset was filtered and only relative abundances higher than 0.01% in at least one replicate were kept in the analyses. This was carried out to eliminate counts that may represent a sequencing artefact and to minimize the number of tests that are unlikely to result in significant findings. Relative abundances greater than null were log-transformed because of the sum constraint [22].

To create the pSEM and identify genera associated or not associated with the weights measured on the day of OTU collection, we used the BayesiaLab (v4.6) software package [44]. Following their recommendations, log-transformed relative abundances were discretized with the R2-GenOpt* algorithm and null relative abundances were grouped into one specific category. Hereafter, these discretized values are called “RA”. To discretize weights, a K-means classification algorithm was chosen after visually inspecting their distribution and they were grouped into five distinct and non-overlapping classes.

Modeling started with the creation of several network structures linking all RAs. To do so, we used all unsupervised learning algorithms proposed in BayesiaLab to increase the probability of finding a solution close to the global optimum. We evaluated the complexity of the networks with respect to the fit to the data via the structural coefficient (SC) analysis of the MDL score [45]. The strength of each relationship in the networks was measured by computing the Kullback–Leibler (KL) divergence between variables included in the relationship [46]. The best network structure was selected as the one with the lowest MDL score and for which all KL values were significantly different from null (G test; *p* < 0.05).

Once this best network was found, we created the measurement part of the pSEM by clustering RAs in accordance with the structure of the network and allowing a maximum number of ten RAs per cluster. The robustness of the resulting clusters was evaluated on ten subsets with the jackknife cross-validation method [47], and the percentages of correspondence of each cluster with respect to the one obtained in the initial network were computed. The quality of the representation of each cluster was measured by its purity (i.e., mean of the RA assignment probabilities in the cluster) and the contingency table fit (which compares the entropy of the current naive Bayes structure to the entropy of a fully connected structure). The LCs were then assigned to each cluster using naive Bayes structures. The value assigned to a LC is the weighted average of all RAs included in the cluster, and the weights are the ratio of the mutual information of the RA with respect to its LC divided by the highest mutual information in the LC.

Finally, the strength of the relationships between the birds’ weight categories, birds’ age at sampling (16, 28, 39, 67 or 73 days), birds’ sex (male or female), origin of the OTU sample (cloaca or caecum), amount of dietary BSF (0%, 5% or 10%) and LCs was assessed in the structural part of the pSEM. The algorithm recommended by the BayesiaLab’s team to construct this part of the pSEM is called the “Structural Priors Learning algorithm”, which is based on a heuristic search of the best network via a bootstrap aggregating approach [44]. Here, again we examined SC and KL values to choose the final model and evaluated its accuracy in estimating weights with the jackknife cross-validation procedure. The results were expressed as the area under the ROC curves, which are plots of the true against the false positive rates in classifying estimated weights. After all validations, total and direct effects of each LC on the weight categories were computed as the change in weights per unit change in the mean of the latent construct before (total effect) and after (direct effect) controlling for the other nodes.

## 3. Results

### 3.1. Descriptive Analytics

Weight means and standard errors are presented in Figure 1: weights were higher in males than females and increased with the age of the birds. After editing, a total of 62 genera were available to create the network and 47 of them were from the phylum *Firmicutes*. Details on relative abundances per genus can be found in Appendix A for each diet (0%, 5%, 10% of insect incorporation), age (16, 28, 39, 67 and 73 days) and sex (male or female) of the birds, and origin of the fecal samples (cloaca or caecum). Mean relative abundances were highest for the *Lactobacillus* genus, with an average of 25.24% and 66.85% in samples taken from the cloaca and caecum, respectively. Besides *Lactobacillus*, fecal samples of the caecum contained on average 11.07%, 8.07%, 6.22% and 6.08% of bacteria of the genera *Lachnospiraceae*, *Ruminococcaceae*, *Alistipes* and *Faecalibacterium*, respectively. In the fecal samples from the cloacal swabs, the mean relative abundance for the genus *Candidatus Arthromicus* was 8.24%. The mean relative abundances of all other bacteria were lower than 5%.

### 3.2. Discretization

Discretization of weights with the K-Means algorithm led to 5 classes: 69.35–169.36 g (14.19%), 169.36–271.03 g (27.10%), 271.03–469.8 g (17.42%), 469.8–698.8 g (24.95%) and 698.8–873.0 g (16.34%). Concerning the RAs, the discretization obtained with the R2-GenOpt* algorithm are provided in Appendix A; in most cases, three RA categories were enough to describe these distributions.

### 3.3. Network Construction

The analysis of the SC curves revealed that SCs between 0.7 and 1 were adequate. All algorithms available in BayesiaLab for the unsupervised learning phase in the construction of the network provided similar MDL values, with the lowest one provided by the EQ algorithm with post-processed Taboo. The resulting network (Figure 2) contained 59 arcs and 62 nodes. Corresponding values for the KL and Pearson’s correlation coefficients are provided in Table 1. All KL values were significantly different from null and most correlations were positive, with the exception of the correlation coefficients between *Lactobacillus* and those of the genera *Alistipes, Parabacteroides* and *Ruminococcaceae NK4A214*, between *Ruminococcaceae UCG-010* and *Alcaligenes* and between *Candidatus Saccharimonas* and *Eschrichia-Shigella*, which were all close to −0.30. Except for two nodes that were too weakly related to the others (i.e., *Clostridium sensu stricto 1* and *7*)*,* most nodes had one incoming and one outcoming arrow. A few ‘dominant’ nodes had more outcoming than incoming arrows: *Lachnospiraceae* (0 in, 6 out), *Ruminiclostridium_5* (1 in, 3 out), *Lactobacillus* (1 in, 5 out), *Ruminococcacceae NKA214* (1 in, 4 out), *Clostridiales vadin BB160* (1 in, 4 out), *Defluvitaleaceae_UCG-011* (1 in, 3 out), *Rombutsoa* (1 in, 3 out), *Tyzzerella* (1 in, 2 out) and *Ruminococcacceae_UCG-010* (1 in, 2 out).

### 3.4. Measurement Part of the pSEM

The RAs were clustered into 11 groups in accordance with the structure of the network shown in Figure 2. Purity and contingency table fit values of the clusters are presented in Table 2. All purity values were higher than 95% (overall mean = 98.27%) and all contingency table fit values (overall mean = 83.97%) were above the recommended threshold of 70%. Cross-validation with the jackknife method showed that nodes are often clustered into the same clusters. The average fit score was 65.40% and ranged from 53.55% to 76.32%.

The characteristics of the 11 LCs associated with these clusters are provided in Appendix A. High LC values usually corresponded to the high relative abundances of their components, with the exception of LC0, for which high values corresponded to high values of *Lactobacillus* and *Candidatus Arthromitus* but low values of the other components.

### 3.5. Structural Part of the pSEM

The structural part of the pSEM is represented in Figure 3 and corresponding metrics are provided in Table 3 for SC = 0.45, which was the network with the lowest structure/data ratio. All KL values were significantly different from null, with the exception of the arc between sex and weight. Results of the ten-fold cross-validation procedure indicated that areas under the ROC curves varied from 92.87% to 96.86% (mean = 95.86%). The structure was characterized by three notable features: the node “weight” is the child node of the nodes “age” and “sex”, the node “age” is the parent of the nodes LC0, LC1, LC4, LC5 and LC6 and the node “BSF” is not linked to the node “weight”. This structure was found in all networks constructed with SC values ranging from 0.45 to 1. It was also found in all of these networks that nodes LC2, LC3, LC7 and LC9 are child nodes of LC6.

Total effects on weights were significantly different from null for LC1, LC4, LC5, LC6, LC8, age and sex: weight increased by 10.59 g per day of age and male birds were heavier than female birds. Among LCs, the strongest total effect was for LC5, with a value of 285.66 (standardized value = 0.57). Total effects of LC1, LC4, LC6 and LC8 were −116.38, 108.96, 492.45 and 380.21, respectively. The other effects were not significantly different from null. Concerning the direct effects on weight, only the effects of age and sex were significantly different from null, with values of 10.52 and 34.50 g for age and sex, respectively.

## 4. Discussion

In this study, it was hypothesized that feeding chicken with different BSF amounts influences their rates of growth via an effect on their gut microbiota. To test this hypothesis, we evaluated the strength of the interrelationships between weight, diet and OTU abundances with a pSEM because this type of model combines the advantages of traditional SEMs with those of Bayesian networks. Age and sex of the birds, as well as the gut compartment where the microbiota was characterized, were also included in the pSEM because these variables may alter the estimated values of the strength of association between weights and microbial abundances.

### 4.1. Descriptive Analytics

Although bacterial relative abundances are known to vary across studies [48], *Firmicutes* was found to be the most abundant phylum and *Lactobacillus* the most abundant genus, as was reported in a recent meta-analysis [49], although in different percentages. The composition of fecal microbiota samples obtained via cloacal swabs or directly from the caecum were different, as observed by others [50,51,52]. Also observed by others, many species could not be classified, encouraging further studies to improve techniques for sequencing and annotating genomes in chicken microbiota [53].

Ardennaise weights (Figure 1) were similar to those found in previous studies: means at 11 weeks of age were 924.70 and 766.51 g for males and females, respectively [54], and 656 g at 8 weeks of age [55].

### 4.2. Measurement Part of the pSEM

Gut bacteria co-exist in complex networks, so the first step in testing our hypothesis was to explore OTU-to-OTU associations and to summarize them into a few representative LCs so the effect of diets on them could be quantified. Results of the pSEM analysis showed that RAs could indeed be combined in distinct unobservable LCs (Appendix A). The composition of these LCs can be compared to the four enterotypes defined by Kaakoush et al. [56]. In their study, they used principal component analysis to aggregate the relative abundances found in 56-day-old chicken feces and discovered 4 enterotypes whose compositions may be related to that of the LCs in this study. For example, enterotype 1 was dominated by *Lactobacillus* (LC0) and *Peptostreptococcaceae* (LC1). Besides genera in enterotype 1, enterotype 2 was dominated by *Streptococcus* (LC1) and all Proteobacteria of LC4 (*Escherichia*, *Shigella* and *Enterobacter*), enterotype 3 was dominated by the only Actinobacteria found in this study (*Corynebacterium 1*) and enterotype 4 was dominated by *Ruminococcaceae* (mainly in LC2, LC3, LC6 and LC7) and Bacteroidetes (*Alistipes* and *Bacteroides,* both in LC0). Conceivably, several of the LCs could have been merged together to better match the composition of these enterotypes, but purity values (>95%) and contingency tables’ fits (>70%) indicated a good representation of the RA joint posterior densities, which is not in favor of further clustering in this study. In another study (layer hens of various ages), Xiao et al. [57] proposed a network of 30 genera connected by 478 links corresponding to absolute Spearman’s rank correlation values between OTU relative abundances higher than 0.5, but available information was not sufficient to support or contradict our findings. The clustering of *Lactobacillus* species with members of the *Bacteroides, Ruminococcaceae* and *Bacteroidales* genera in LC0 was similarly observed by Zou et al. [58], and this is another argument in favor of the structure of our network.

Metabolic dependencies among bacteria may partly explain the composition of these LCs [59]. Indeed, cecal microbiota plays important roles in the metabolic pathways leading to the production of short-chain fatty acids [60,61] and compounds from the fermentation of ileal bypass proteins [62]. For example, RAs of *Lactobacilli* (LC0) were inversely associated with RAs of many other genera (Table 1), and this may be substantiated by the ability of *Lactobacilli* to produce bacteriocin-like substances potentially toxic for other bacteria and to convert carbohydrates into lactic acid, which decreases the pH, which usually restricts bacterial growth [48]. Similarly, bacteria belonging to four out of the six genera of LC3 (i.e., *Oscillibacter*, *Lachnospiraceae Ruminococcacea* and *Subdoligranulum*, all with high loadings) were shown to be involved in butyrate synthesis [63,64,65]. Bacteria belonging to four out of the five genera of LC5 (i.e., *Megamonas*, *Phascolarctobacterium*, *Barnesiella* and *Parabacteroides*) expressed enzymes for propionate production pathways [64]. Members of *Alistipes* and *Bacteroides*, both with high loadings in LC0, were shown to convert carbohydrates, including cellulose, through the succinate pathway [66,67]. Relative abundances of *Eisenbergiella* and *Ruminococcaceae UCG-014*, the only components of LC10, were positively correlated with the amount of butyric acid in the caecum of chicks infected with the porcine delta-coronavirus [68]. In humans, *Shuttleworthia*, with the highest loading in LC7, mainly produce butyrate via fermentation of non-digestible carbohydrates [69] and were associated with male broilers of high body weights [70].

Besides carbohydrates, bacteria may also use proteins as fermentable substrates, as do bacteria of genera *Peptostreptococcaceae*, *Romboutsia* and *Paraclostridium* [71,72] with the highest abundances among the OTUs of LC1. Bacteria also play a role in chicken immunity. For example, LC2 is dominated by bacteria of *Clostridiales vadinBB60* and *Mollicutes RF39* genera. Both were positively correlated with levels of regulatory T cells and CD11C^+^CD103^+^ dendritic cells in mesenteric lymph nodes of mice sensitized with ovalbumin [70]. Similarly, *Candidatus Arthromitus* and *Bacteroides* (both in LC0) have been shown to activate the innate and adaptive immunity [56,73,74] and *Eisenbergiella* (LC10) was negatively correlated with TNF-α and IFN-γ expression levels in chicks infected with the porcine delta-coronavirus [68].

### 4.3. Structural Part of the pSEM

The second step to test our hypothesis was to explore links across the LCs and diet, while controlling for variables such as sex, age and gut compartment, which may also influence weights. We observed that constructs LC2, LC3, LC7, LC9 and LC10 were child nodes of LC6 in all networks created with different SC values. Nutritional interactions may explain these relationships because metabolic cross-feeding is frequent in these microbial communities, as was reviewed by Flint et al. [75]. One may then imagine that bacteria of LC6 convert substrates into products subsequently used by bacteria of LC2, LC3, LC7, LC9 and LC10, but further analyses are necessary to validate or reject this hypothesis.

Age was shown to influence body weights and microbiotal composition in this (Figure 3 and Table 3) and many other studies [7,8,76,77]. The KL values were highest between the “age” node and the nodes of “weight”, LC1 and LC4 (Table 3). In support of this observation, the genus *Faecalibacterium*, the most abundant genus of LC4 (with relative abundances going from 0.06% at 16 days of age to 6.95% at 73 days of age) was previously associated with mature microbiota in humans and chicken [78]. Richards et al. [4] also observed that the genus *Faecalibacterium* colonizes the cecum later in life of broilers of three different breeds, as well as other members of the *Ruminococcaceae*, *Firmicutes* and *Mollicutes RF39,* but this last observation was not supported in our study. A link also connected the nodes “age” and “BSF”, which can be explained by the design of the study. Indeed, birds at ages 16 and 28 days were sampled before BSF larvae were included in their diets, while older birds were fed on a diet supplemented or not with BSF larvae.

No link was found between the node “BSF” and any of the LCs. Similarly, little or no differences were found in the composition of cecal microbiota of Ross broilers fed a commercial feed with either 0% or 8% of fresh BSF larvae [13], or up to 20% [79], while others observed differences in the relative abundances of several genera in Ross broilers fed meals including 0% to 15% of partially defatted BSF larvae [10] or meal in which soybean oil was partially substituted by natural or modified BSF larvae fat [80]. It is possible that the substances in BSF fat explain these findings. Indeed, fat of BSF larvae is rich in medium-chain fatty acids known to influence microbiota and intestinal villi morphology [15,80,81].

Likewise, no link was created between the node “BSF” and the node “weight”. This suggests that BSF larvae incorporation in the diets of broilers would not affect the weight of birds of a similar age. This was found by many, but not all studies, as reviewed by Abd El-Hack [82]. Discrepancies were also observed in meta-analyses of the growth of chicken [2] and fish [82] fed or not fed BSF larvae. No main effect was observed in both meta-analyses, but replaced protein sources and the BSF inclusion level were found responsible for the discrepancies between the studies included in the meta-analyses. Rearing substrates, environmental conditions and harvesting and processing methods of BSF larvae were also shown to explain the differences between studies [83,84]. In this study, soybean meal was replaced by up to 5% of fresh BSF larvae and diets were formulated to have equal energy and protein concentrations. The low incorporation rates of the BSF larvae [2] and the similarity of the amino acid profiles of these larvae and soybean meal [85] could explain the small difference observed between the weight means of birds fed meal supplemented or not with BSF larvae. On the other hand, the growth rate was shown to be reduced when broilers were fed higher levels of BSF larvae than those used in this study [2,16,86].

Finally, we observed that the direct effects on weights of all LCs were null, while the total effects of LC1, LC4 and LC6 were different from null—negative for LC1 and positive for LC4 and LC6. Therefore, because the node “age” was intermediate within the pathway linking the nodes for these LCs and the node “weight” (Figure 3), it can be deduced that LC1 indirectly decreased and LC4 and LC6 indirectly increased body weights via their effect on age. These findings point to the importance of mediation analysis when exploring the mechanisms underlying the relationships between microbiota, age and weight.

### 4.4. Limitations

The proposed methodology obviously has several limitations, which should be taken into account in the interpretation of our data. For example, factors such as the genetic background, immune system activity and behavior of the birds were not considered, although they are known to influence microbiotal composition [87]. Feed intake was not measured and may have affected growth performances. However, results reported in the literature are conflicting [16]. For example, feed consumption was not statistically different between Ardennaise chickens fed a commercial diet, of which 8% was replaced by fresh BSF larvae, and chickens fed the control diet [88]. The environment was kept as constant as possible across replications, but it is still possible that the effect of age on some LCs is rather due to environmental conditions than age by itself, especially since the effects of age, location and replication are partially confounded in this study. It may also be argued that the sample size was small and limits the statistical power of the network. On the other hand, the MDL principle that we used is known to be particularly relevant for small samples [89], although it is possible that more significant differences would be observed with a larger sample size.

Discretization of the observed relative abundances into RAs yielded only a few classes, which may be interpreted as a loss of information, but Lehmann and Hurlbert [90] observed that two and three ordered categories are often appropriate when discretizing continuous data. Additionally, discretization allows evaluating non-linear relationships [91]. This is particularly relevant in this study as quite a few relationships were non-linear, as suggested by the differences between KL divergence and Pearson’s correlation coefficients (Table 1 and Table 3). Lastly, results of the cross-validation procedures provided high values for the areas under the ROC curves, higher than the threshold of 80%, suggesting good classification performance [92].

## 5. Conclusions

In this study, we have developed a new method to assess the direct and indirect roles of gut bacterial microbiota in the growth of chickens fed or not fed BSF larvae. In this first application, it was shown that the pSEM reduced the complex web of relationships among the identified bacterial genera into a few unobservable latent constructs, which showed distinctive metabolic attributes and nutritional dependencies. Direct links were found between two of these constructs, one containing the genus *Faecalibacterium*, with the birds’ age and weight. No direct link was found between the level of BSF supplementation and growth. Application of this method to microbiota data from other sources may provide valuable insight into how it may influence chickens’ growth.

These results collectively provide an improved understanding, although further studies are necessary to validate causal associations.

## Figures and Tables

**Figure 1 biology-11-00357-f001:**
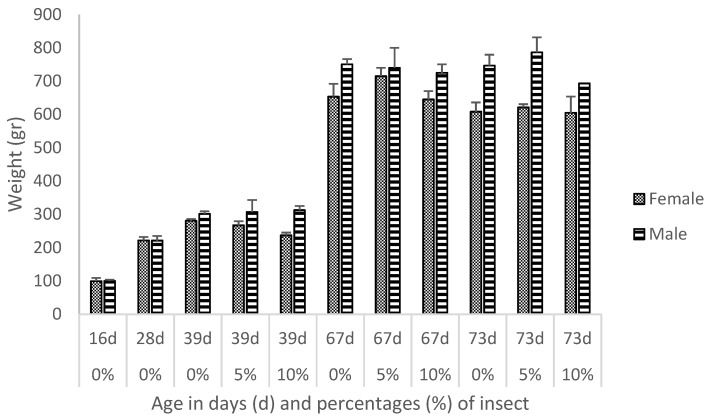
Weight means and standard errors (g) per sex, age of the birds in days (d) and percentages of *Hermitia Illucens* (%) in the diets.

**Figure 2 biology-11-00357-f002:**
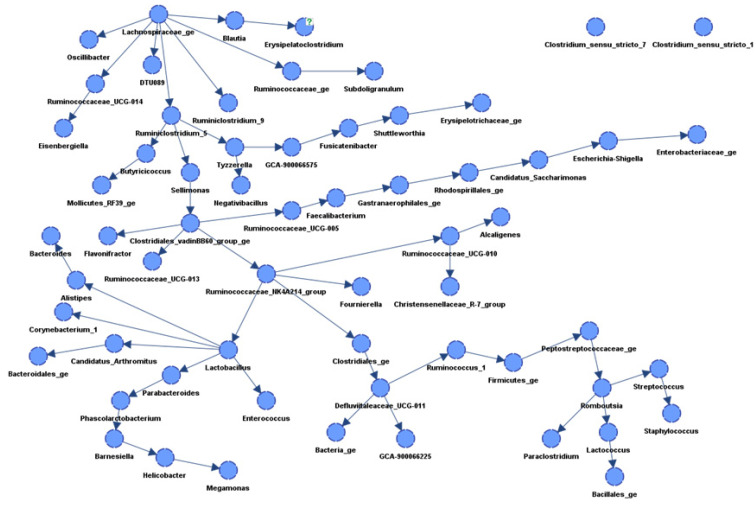
Network showing the arcs linking the bacterial relative abundances.

**Figure 3 biology-11-00357-f003:**
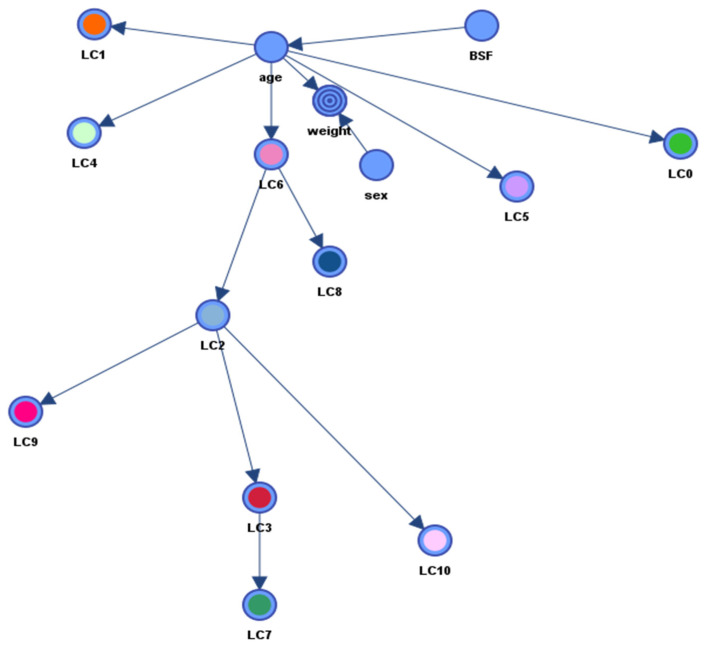
Network showing arcs between weight, latent constructs (LC0 to LC10), age, sex and BSF nodes.

**Table 1 biology-11-00357-t001:** Relationship analysis of the network depicted in Figure 1: Kullback–Leibler divergence (KL) and Pearson’s correlation (Corr) between parent and child nodes.

Parent Node	Child Node	KL	Corr
*Ruminococcaceae NK4A214 group*	*Ruminococcaceae UCG-010*	0.77	0.83
*Lachnospiraceae*	*Ruminococcaceae*	0.76	0.94
*Ruminiclostridium 5*	*Butyricicoccus*	0.76	0.88
*Tyzzerella*	*Negativibacillus*	0.75	0.86
*Ruminococcaceae UCG-010*	*Christensenellaceae R-7 group*	0.73	0.81
*Fusicatenibacter*	*Shuttleworthia*	0.73	0.71
*Tyzzerella*	*GCA-900066575*	0.72	0.85
*Clostridiales vadinBB60 group*	*Flavonifractor*	0.72	0.76
*Clostridiales vadinBB60 group*	*Ruminococcaceae NK4A214 group*	0.72	0.79
*Ruminiclostridium 5*	*Sellimonas*	0.69	0.83
*Ruminococcaceae UCG-014*	*Eisenbergiella*	0.69	0.84
*GCA-900066575*	*Fusicatenibacter*	0.68	0.83
*Ruminiclostridium 5*	*Tyzzerella*	0.68	0.82
*Lachnospiraceae*	*Ruminococcaceae UCG-014*	0.67	0.88
*Lachnospiraceae*	*Ruminiclostridium 5*	0.67	0.86
*Lachnospiraceae*	*Ruminiclostridium 9*	0.66	0.86
*Ruminococcaceae NK4A214 group*	*Clostridiales*	0.65	0.77
*Lachnospiraceae*	*Oscillibacter*	0.65	0.84
*Clostridiales*	*Defluviitaleaceae UCG-011*	0.65	0.71
*Escherichia-Shigella*	*Enterobacteriaceae*	0.63	0.73
*Sellimonas*	*Clostridiales vadinBB60 group*	0.62	0.75
*Ruminococcaceae*	*Subdoligranulum*	0.61	0.85
*Alistipes*	*Bacteroides*	0.59	0.69
*Clostridiales vadinBB60 group*	*Ruminococcaceae UCG-013*	0.58	0.70
*Ruminococcaceae NK4A214 group*	*Lactobacillus*	0.57	−0.81
*Ruminococcaceae UCG-005*	*Faecalibacterium*	0.57	0.78
*Lachnospiraceae*	*DTU089*	0.56	0.78
*Rhodospirillales*	*Candidatus Saccharimonas*	0.54	0.58
*Defluviitaleaceae UCG-011*	*GCA-900066225*	0.54	0.70
*Lactobacillus*	*Alistipes*	0.54	−0.74
*Shuttleworthia*	*Erysipelotrichaceae*	0.53	0.64
*Clostridiales vadinBB60 group*	*Ruminococcaceae UCG-005*	0.51	0.72
*Lachnospiraceae*	*Blautia*	0.48	0.57
*Barnesiella*	*Helicobacter*	0.48	0.63
*Ruminococcus 1*	*Firmicutes*	0.48	0.26
*Firmicutes*	*Peptostreptococcaceae*	0.46	0.36
*Lactobacillus*	*Candidatus Arthromitus*	0.45	0.75
*Faecalibacterium*	*Gastranaerophilales*	0.45	0.62
*Defluviitaleaceae UCG-011*	*Ruminococcus 1*	0.45	0.69
*Butyricicoccus*	*Mollicutes RF39*	0.44	0.66
*Ruminococcaceae NK4A214 group*	*Fournierella*	0.43	0.51
*Defluviitaleaceae UCG-011*	*Bacteria*	0.40	0.53
*Romboutsia*	*Paraclostridium*	0.39	0.53
*Peptostreptococcaceae*	*Romboutsia*	0.39	0.73
*Blautia*	*Erysipelatoclostridium*	0.38	0.59
*Helicobacter*	*Megamonas*	0.36	0.68
*Romboutsia*	*Lactococcus*	0.36	0.64
*Lactobacillus*	*Corynebacterium 1*	0.36	0.28
*Parabacteroides*	*Phascolarctobacterium*	0.35	0.64
*Lactobacillus*	*Enterococcus*	0.33	0.39
*Lactococcus*	*Bacillales*	0.33	0.63
*Gastranaerophilales*	*Rhodospirillales*	0.32	0.52
*Ruminococcaceae UCG-010*	*Alcaligenes*	0.30	−0.34
*Streptococcus*	*Staphylococcus*	0.30	0.62
*Romboutsia*	*Streptococcus*	0.30	0.42
*Phascolarctobacterium*	*Barnesiella*	0.27	−0.02
*Candidatus Saccharimonas*	*Escherichia-Shigella*	0.27	−0.30
*Lactobacillus*	*Parabacteroides*	0.23	−0.36
*Candidatus Arthromitus*	*Bacteroidales*	0.14	−0.24

**Table 2 biology-11-00357-t002:** Mean purity and contingency table fit (CTF) of each cluster.

Cluster	Purity (%)	CTF (%)
0	99.65	79.32
1	98.91	61.79
2	98.46	80.41
3	98.23	88.48
4	97.98	76.14
5	99.57	80.59
6	98.98	87.11
7	98.66	87.84
8	96.82	87.85
9	97.28	96.10
10	96.43	98.02

**Table 3 biology-11-00357-t003:** Relationship analysis of the network depicted in Figure 2: Kullback–Leibler (KL) divergence and Pearson’s correlation between parent and child nodes.

Parent Node	Child Node	KL Divergence	Correlation
age	weight	1.51	0.94
age	LC1	1.34	−0.24
LC6	LC2	0.95	0.62
age	LC4	0.92	0.53
LC6	LC8	0.92	0.56
LC2	LC10	0.87	0.84
LC2	LC3	0.82	0.86
age	LC6	0.78	0.37
LC3	LC7	0.78	0.79
age	LC0	0.68	−0.70
age	LC5	0.68	0.53
LC2	LC9	0.41	0.68
BSF	age	0.34	0.44
sex	weight	0.18	0.07

## Data Availability

The data presented in this study are available within the article and the Appendix A.

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
