# Peer review of "A Probabilistic Structural Equation Model to Evaluate Links between Gut Microbiota and Body Weights of Chicken Fed or Not Fed Insect Larvae"

_biology, 2022, doi:10.3390/biology11030357_

Round 1

Reviewer 1 Report

It can be accepted in the current version.

Author Response

Thank you for your efforts in improving the manuscript

Reviewer 2 Report

Suggested changes have not been fully implemented. Despite that, the manuscript has been improved.

Author Response

Thank you for your efforts in improving the manuscript

This manuscript is a resubmission of an earlier submission. The following is a list of the peer review reports and author responses from that submission.

Round 1

Reviewer 1 Report

The author’s experimental design bases the hypthosize that feeding chicken with black soldier fly larvae (BSF) may influence their rates of growth via effects on the composition of their gut microbiota. However the impact of diets supplemented with BSF on growth performance hasn't discussed (or few) in the whole manuscript. Only found age that has an effect on growth performance in chicken, then what's the meaning of this experiment? In addition, the information obtained from Table 1 seems to support our doubt.

In order to display the results more intuitively, it is recommended that the author change Table 1 to a picture. Something wrong exist in Table 3.

Generally speaking, feed intake should be consider when evaluating the growth performance. The author did not record this work. How to exclude the impact of feed intake on growth performance? You know, diets supplemented with BSF may change the palatability of diets.

There are language errors in the manuscript, and the author is recommended to check it carefully.

Reviewer 2 Report

General comments

L43-56: the authors should elaborate more on the nutrients contained in the black soldier fly that could be unique for poultry and probably influence the gut microbiota composition and/or histomorphometry of the gut. This paragraph needs to be revised.

L319-370: what is the relationship of this part regarding the BSF (%)? It is not easy to the reader to distinguish hoe these results are related to the level of black soldier fly in the ration. This

As it is discussed in few lines (L397-398) “This suggests BSF  larvae incorporation in the diets would not affect the weight of birds of similar age”, then the title of the manuscript “evaluate links between gut microbiota and body weights of local chicken fed in sect larvae” does not resemble the main findings of the study. The authors should reconsider the way data are presented and maybe the title of the manuscript.

Minor comments

L9: change would to could

L14: insert were before probably

L20: it would be more suitable to change “death” to slaughter. Also add the age at slaughter.

L21: specify at what age the birds were weighed

L131: duration of the experimental period

L134: a brief description could be provided at this point as well.

Tables 1,2,3: should be formatted according to the guidelines of the journal

L293: also format Table 5, but where is Table 4.